# Pulmonary Metastasizing Low-Grade Endometrial Stromal Sarcoma: Case Report and Review of Diagnostic Pitfalls

**DOI:** 10.3390/diagnostics12020271

**Published:** 2022-01-21

**Authors:** Geon Woo Kim, Sun Kyung Baek, Jae Joon Han, Hong Jun Kim, Ji-Youn Sung, Chi Hoon Maeng

**Affiliations:** 1Division of Medical Oncology-Hematology, Department of Medicine, College of Medicine, Kyung Hee University Hospital, Seoul 02447, Korea; khkyj16@khmc.or.kr (G.W.K.); wkiki@khu.ac.kr (S.K.B.); anemia@khu.ac.kr (J.J.H.); xpassion84@khu.ac.kr (H.J.K.); 2Department of Pathology, Kyung Hee University Hospital, Kyung Hee University College of Medicine, Seoul 02447, Korea; jysung@khu.ac.kr

**Keywords:** benign metastasizing leiomyoma, endometrial stromal tumor, letrozole

## Abstract

Pulmonary manifestations of benign metastasizing leiomyoma (BML) usually include multiple well-defined, round, bilateral nodules. Low-grade endometrial stromal sarcoma (LG-ESS) is a rare uterine tumor. A 70-year-old woman visited the clinic complaining of acute cough and dyspnea in April 2017. Chest computed tomography (CT) revealed pneumothorax and multiple pulmonary nodules. She had a history of hysterectomy for uterine leiomyoma 23 years ago. Biopsy revealed that the pulmonary masses were consistent with BML. However, the patient had two subsequent episodes of acute, recurrent respiratory distress, accompanied by massive pleural effusions and hydropneumothorax over the next two years. A chest CT performed for acute dyspnea revealed large and multiple hydropneumothoraces. The size and distribution of pulmonary masses were aggravated along with cystic changes and bilateral pleural effusions. Given this aggressive feature, additional immunohistochemical findings and gynecologic pathologist review confirmed the correct diagnosis to be LG-ESS. After initiating anti-estrogen therapy, the patient achieved a partial response, without recurrence of symptoms, for 28 months. Metastatic LG-ESS responds well to anti-hormonal therapy. If the clinical pattern of a disease is different than expected, the possibility of a correction in the diagnosis should be considered.

## 1. Introduction

Benign metastasizing leiomyoma (BML) is a very rare condition that usually affects women of reproductive age with a history of uterine leiomyoma (LM) resection or hysterectomy [1,2]. It is characterized by a well-differentiated and benign smooth muscle tumor that metastasizes to the extrauterine region. Despite the term “metastasizing” in the name of the disease, uterine LM is a benign tumor. Thus, BML does not exhibit malignant potential and related features, such as invasiveness, tissue destruction, aggressiveness, and high mitotic rate [3]. More than half of BML patients have no specific symptoms [4]. Although metastases can occur anywhere, the lungs are the predominant sites of metastasis. It has been reported that concomitant metastasis to the lungs and other sites is rare [5]. The natural course of the disease is generally indolent and requires no active treatment. Therefore, many patients are incidentally diagnosed with radiological imaging during routine medical screening [6]. Pulmonary manifestations of BML usually include radiographic features, such as multiple well-defined, round, and bilateral nodules of various sizes [4]. Cystic changes or cavitary features with occasional pneumothorax have also been rarely reported [6,7,8,9].

Low-grade endometrial stromal sarcoma (LG-ESS), a subtype of endometrial stromal tumors, is a very rare tumor of low malignant potential and is characterized by the proliferation of short spindle cells with low nuclear atypia and low mitotic index [10]. LG-ESS has an indolent clinical course, although late recurrence or distant metastasis may occur after curative surgery [11,12]. Since ESS can be misdiagnosed as LM, differential diagnosis between BML and ESS is often challenging [11,12]. We recently examined a case of symptomatic and progressive BML presenting with massive pleural effusion, hydropneumothorax, bulky multiple lung masses with cavitary changes, and extrapulmonary metastasis—a mixture of various clinical features reported to date. In contrast to the known clinical features of BML, those in this case were atypical and aggressive. Therefore, after requesting a review of the case by a gynecologic pathologist, we concluded that tumors in this patient should more accurately be classified as LG-ESS. This case report emphasizes the importance of attention to the clinical course, and clinician involvement to obtain accurate diagnosis to ensure patient improvement.

## 2. Case Description

### 2.1. Initial Presentation

A 70-year-old woman visited the clinic complaining of acute cough and dyspnea in April 2017. A chest X-ray and chest computed tomography (CT) revealed pneumothorax in the left chest and multiple nodules in both the lungs (Figure 1).

The patient had been diagnosed with bilateral lung nodules in 2002 (15 years prior) at another hospital and was aware of the lung lesions. The patient was closely followed up without treatment since diagnosis, as no active symptoms were noted. However, regular follow-up appointments were discontinued by the patient 7 years ago. She had undergone bilateral salpingo-oophorectomy with hysterectomy due to uterine myoma in 1994 (23 years prior). Comparison of previous chest CT performed outside the hospital between 2002 and 2008 revealed that the size and number of pulmonary nodules had slowly increased, although the patient did not complain of any symptoms during this period. The body mass index of the patient was 20.9 kg/m^2^ (height 152 cm, body weight 48.5 kg). She has not received any type of estrogens such as hormonal therapy. No other special medical history or medication was recorded. Video-assisted thoracoscopic surgery was performed for the treatment of pneumothorax, and tissue biopsy was performed to confirm the diagnosis of lung nodules. Based on the patient’s medical history and serial review of radiological findings, the histological diagnosis was BML. Abdominal CT, which was performed to determine the extent of the disease, revealed other masses in the pelvic cavity. However, the pelvic mass was asymptomatic. Excisional biopsy of the pelvic mass was also performed after stabilization of the pneumothorax. Similar to the pulmonary masses, the pelvic mass was also diagnosed as BML. Immunohistochemical staining was positive for estrogen receptor (ER) and progesterone receptor (PR) at both tissue sites. Symptoms improved after resolution of pneumothorax, leading to the decision to regularly follow-up the patient without further treatment.

### 2.2. Disease Progression and Revision of Prior Diagnosis

Subsequently, the patient experienced two more episodes of recurrent dyspnea owing to increased pleural effusion within the two years following the initial onset of symptoms. A chest CT performed at the time of the third hospitalization for acute dyspnea in June 2019 revealed large and multiple hydropneumothoraces and bullae with massive bilateral pleural effusions (Figure 2).

The size and distribution of the pulmonary masses were aggravated along with cystic changes. Considering the rapid growth in a relatively short time and associated distressing symptoms, the decision was made to initiate active antitumor treatment instead of continuing with conservative management and close observation. The patient had previously undergone bilateral salpingo-oophorectomy and was menopausal. Therefore, letrozole, an aromatase inhibitor, was initiated at 2.5 mg once daily. At this time, considering the patient’s aggressive and rapid clinical progression, which was not considered benign, the possibility of malignancy, rather than BML, was raised. Therefore, it was decided to review the surgical specimens of the lung and pelvic masses from 2017. Microscopic examination revealed that the metastatic lung nodules were small, uniform, characterized by bland tumor cells with spindled nuclei and scant cytoplasm, in the background of rich small arterioles or capillary networks (Figure 3). Mitotic counts were up to 1 per high-power field. Necrosis was not present.

Immunohistochemical (IHC) staining showed that desmin was expressed in some tumor cells. The tumor cells showed patchy cytoplasmic immunoreactivity with CD10, while additional immunohistochemical staining revealed that caldesmon was diffusely positive in the tumor cells. Diffuse nuclear positivity of tumor cells with Wilms’ tumor-1 (WT-1) was also noted (Figure 4). Based on a thorough review by a gynecologic pathologist, the tissue was found to be more suitably categorized as LG-ESS, rather than BML (Figure 4). Because our hospital does not perform suitable molecular tests for genetic fusions of LG-ESS, such as *JAZF1* translocation, molecular alterations could not be evaluated.

### 2.3. Treatment and Clinical Course

After initiating hormonal therapy, the size and number of lung masses decreased on subsequent chest CT, and bilateral pleural effusions disappeared completely. The size of the pelvic masses also decreased slightly but not to the extent observed in lung lesions. Although tumors were still unresectable, patient condition has been stable without significant toxicity from the treatment or recurrence of symptoms for 28 months (Figure 5). Her treatment is ongoing.

## 3. Discussion

The most common site for BML is the lungs [3]. Since BML is not malignant and is composed of benign smooth muscle cells, vascular spread can occur due to blood vessel injury during gynecological surgery, rather than spontaneous vascular invasion—one of the hallmarks of malignancy. It is well known that there is a higher prevalence of BML among women who undergo myomectomy or hysterectomy [3]. Seeding or implantation of fragmented leiomyomatous cells during a surgical procedure is another hypothesis, especially in cases of BML in the abdominopelvic cavity [3]. The present case was initially diagnosed with BML. Therefore, until the second event of acute respiratory distress, systemic anti-cancer treatments were not considered. However, as the patient’s clinical course showed atypical and aggressive features compared with that observed in usual cases of BML, a revised and correct diagnosis was obtained after gynecologic pathologist re-diagnosis.

The reasons for misdiagnosis of BML in this case can be summarized as follows: (1) Pulmonary BML was initially suggested based on patient’s gynecological history. Review of old medical records from another hospital (from 27 years ago) indicated that the reason for the patient’s hysterectomy was bulky uterine LM; (2) Since the discovery of lung masses in 2002, the patient had no symptoms for more than 15 years and had a very indolent course, thus excluding the possibility of malignancy; (3) Histology obtained from both lung and pelvic masses indicated BML diagnosis. Correct diagnosis would benefit from re-review of surgical specimens from the hysterectomy performed 23 years ago. Unfortunately, this was impossible due to specimens being obtained too long ago, and the patient underwent the operation in another hospital. It is possible that the hysterectomy specimens of the patient would be consistent with LG-ESS, if they could be re-reviewed. It has already been reported that the histological diagnosis of the primary lesion was changed to LG-ESS retrospectively based on pulmonary metastatic lesions after several decades (Table 1). In contrast, the following reasons were considered for why BML was less likely in this case: (1) Our patient had pulmonary and pelvic lesions. Simultaneous involvement of the lungs and non-pulmonary sites is particularly rare among cases of BML reported to date [5]; (2) The disease progressed despite the patient being menopausal and having undergone bilateral salpingo-oophorectomy along with hysterectomy. Since BML in premenopausal women usually tends to disappear spontaneously after menopause, the clinical course of this patient was rather peculiar and aggressive, given that she had severe and recurrent respiratory distress with various clinical features, despite having been menopausal for long time; (3) Pleural involvement is a very rare feature of BML [4]. Our patient experienced recurrent episodes of acute respiratory distress caused by massive pleural effusion (hydropneumothorax) and bulky masses. Pleural fluid analysis using multiple thoracentesis showed that the fluid was consistently an exudate (protein level: 4.1–4.8 g/dL). Although cytology of the pleural fluid was negative for tumor cells and pleural seeding was not evident on serial chest CT, numerous nodules were located adjacent to the pleura, suggesting possible involvement of the visceral pleura.

Because of these complexities, the clinician’s rational suspicion for the diagnosis, and expert pathological evaluation by an experienced gynecologic pathologist were essential for the accurate diagnosis of LG-ESS [13]. From a pathological point of view, the differential diagnosis of LM and LG-ESS is a bewildering issue of uterine mesenchymal tumors. In particular, when pathologists observe metastatic tumors without knowing the morphology of primary uterine lesions, as in our case, reliable diagnosis is difficult. There are three points for diagnostic pitfalls of LG-ESS. First, this disease group has been recently redefined and well established. In fact, since endometrial stromal tumors were sub-categorized into four groups in 2014, these tumors were re-classified by the World Health Organization in 2020. Therefore, patients with a history of uterine mesenchymal tumor, such as LM, diagnosed long ago, may be re-diagnosed due to different diagnostic terms at that time, even if it is LG-ESS based on the current criteria. Second, there is minimal cellular atypia in LG-ESS. Without a thorough examination of the characteristic morphology of LG-ESS, it is very easy to misdiagnose LM when the pathologist evaluates the benign-looking uterine mesenchymal tumor. Third, LG-ESS can be positive for desmin and h-caldesmon and typically highlights smooth muscle differentiation [14]. Although immunohistochemical antibodies against smooth muscle differentiation markers, such as desmin and h-caldesmon (positive for LM and negative for LG-ESS), and endometrial stromal tissue markers, such as CD10, vimentin, and WT-1 (positive for LG-ESS and negative for LM), can be useful, misdiagnosis can still occur if clinicians do not consider the above exceptions.

**Table 1 diagnostics-12-00271-t001:** Summary of published literature reporting pulmonary metastasis from LG-ESS.

Case No.	Age (Years)	Clinical Manifestations	Other Metastases	RFS (Years)	Radiological Findings	Treatment	Follow-Up	Reference
1	37	NR	Pelvis	0.8	NR	Chemotherapy with radiotherapy	AWD	[12]
2	48	NR	Vagina	8	NR	Chemotherapy	AWD	
3	58	NR	Bone	15.5	NR	NR	AWD	[15]
4	42	Dyspnea, tachycardia	Heart	6	Multiple lung masses	Surgery	AWD	[16]
5	59	NR	None	0.6	NR	Chemotherapy (gemcitabine, docetaxel) followed by MPA	AWD	[17]
6	43	NR	Pelvis	4.2	NR	Chemotherapy (ifosfamide, carboplatin, doxorubicin) followed by MPA	AWD	
7	58	NR	Pelvis, LN	1.8	NR	MPA followed by chemotherapy, radiation, and letrozole	AWD	
8	32	NR	None	6.8	NR	MPA	DOD	
9	41	NR	None	10	NR	MPA	AWD	
10	68	NR	None	23	Solitary mass	Surgery	NR	[18]
11	58	NR	Para-aortic LN	1.6	Multiple lung masses	MPA and chemotherapy followed by letrozole	AWD	[19]
12	44	Asthenia, weight loss	Rectum	0.3	Multiple lung masses	Aminoglutethimide and hydrocortisone	Alive (CR)	[20]
13	34	NR	None	1	Multiple lung masses	Aminoglutethimide and hydrocortisone	Alive (CR)	
14	58	Pneumothorax	None	16	Multiple thin-walled cysts	NR	NR	[21]
15	45	Dry cough	None	25	Multiple lung masses	None (the patient refused hormonal therapy)	DOD	[22]
16	51	NR	Pelvis	11	Multiple lung masses	Hormonal therapy followed by surgery	Alive (CR)	[23]
17	56	Right clavicle pain	None	5	Multiple lung masses	Surgery followed by hormonal therapy	Alive (CR)	
18	38	Incidental	None	5	Solitary mass	Surgery	Alive (CR)	
19	31	Incidental	None	2.5	Multiple masses with cystic changes	Surgery	Alive (CR)	
20	77	Incidental	None	13	Multiple lung masses	Surgery	NR	
21	46	Dyspnea, cough, chest pain	None	16	Pleural effusion	Hormonal therapy	AWD	
22	48	Dyspnea, cough, chest pain	None	20	Multiple lung masses with pleural effusion	Surgery	DOD	
23	40	Dyspnea, chest pain	None	3	Bilateral reticulonodular infiltrates	Hormonal therapy	AWD	
24	55	Incidental	None	7	Multiple lung masses	Hormonal therapy	AWD	
25	43	RLQ pain	None	7	Solitary mass	Hormonal therapy followed by surgery	Alive (CR)	
26	46	NR	Pelvis	15	Multiple lung masses	Surgery	Alive (CR)	
27	53 ^(a)^	Dyspnea, cough, chest pain	None	10	Multiple lung masses	Chemoradiotherapy	Alive (CR)	
28	32 ^(a)^	Pneumothorax	None	3	Pleural thickening with cystic mass	None	AWD	
29	67 ^(a)^	Incidental	None	9	Solitary mass	Surgery	Alive (CR)	
30	67 ^(a)^	Incidental	None	8	Solitary mass	Surgery	AWD	
31	53 ^(a)^	Incidental	None	4	Multiple mass with cystic change	Surgery	NR	

Cases no. 27–31 designated as ^(a)^ patients not diagnosed with LG-ESS at the time of initial diagnosis. They had a revised diagnosis of LG-ESS after relapse. The initial diagnosis of each case was as follows: case no. 27, leiomyosarcoma; case no. 28, epithelioid leiomyoma; case no. 29, sex-cord stromal tumor; case no. 30, cystic hyperplasia; case no. 31, epithelioid leiomyoma. Abbreviations: LG-ESS, low-grade endometrial stromal sarcoma; RFS, relapse-free survival; y, year; NR, not reported; LN, lymph node; MPA, medroxyprogesterone acetate; AWD, alive with disease; DOD, die of disease; CR, complete response; RLQ, right lower quadrant.

Because none of the above immunohistochemical markers are specific, and their expression patterns vary (diffuse, patchy, or negative), several cases would be difficult to evaluate by IHC staining alone. Our case also showed patchy positivity for desmin, diffuse positivity for h-caldesmon, patchy positivity for CD10, and diffuse positivity for WT-1. Ultimately, the key criterion to differential diagnosis between LM and LG-ESS is morphology [24]. Typical features, including a rich delicate network of small arterioles and capillaries and tumor cells showing swirling patterns around these vessels, are compatible with LG-ESS [25]. When a pathologist observes a low-grade uterine mesenchymal tumor, the possibility of LG-ESS should include a differential diagnosis and focus on its typical features. LG-ESS is commonly misdiagnosed, but high-grade ESS (HG-ESS) should also be considered an alternative diagnosis for BML. Since HG-ESS may contain an LG-ESS area, the whole area should be carefully examined; additionally, the observation of necrosis and increased mitosis favors an HG-ESS diagnosis. Finally, both LG-ESS and HG-ESS are known to have characteristic genetic changes (for LG-ESS, fusions of *JAZF1-SUZ12*, *JAZF1-PHF1*; for HG-ESS, fusions of *YWHAE-NUTM2, ZC3H7B-BCOR*); hence, it would be helpful to conduct a molecular test, if possible [26,27,28].

No standard treatment has yet been established for LG-ESS. Radiotherapy and surgical excision can be considered even when metastatic lesions are resectable [10].

Since proliferation and differentiation of endometrial stromal cells are regulated by estrogen and progesterone, hormonal therapy has been a mainstay for recurrent or metastatic tumors. ER and PR were expressed in up to 80% and 90% of LG-ESS cases, respectively [29]. Progestins, such as medroxyprogesterone acetate (MPA) and megestrol acetate, have been widely used as therapeutic modalities [30]. Aromatase inhibitors, including exemestane (25 mg/day), anastrozole (1 mg/day,) and letrozole (2.5 mg/day), are preferred therapeutic options. A number of case reports revealed that hormonal therapy yielded a good, durable response, or even complete response with acceptable long-term survival in patients with metastatic disease [20,23,30,31]. Gonadotropin-releasing hormone (GnRH) analogs can also be considered, especially in premenopausal patients, because these agents suppress ovarian estrogen synthesis. GnRH analogs can be administered concomitantly with aromatase inhibitors and are recommended in cases of negative ER with positive PR [30]. Rarely, in cases of both ER/PR negative, the effect of hormonal therapy is unclear. Some authors have reported treatments with cytotoxic chemotherapy, such as gemcitabine plus docetaxel or doxorubicin plus ifosfamide, which have often been used for soft tissue sarcomas for metastatic LG-ESS regardless of hormone receptor status. The authors reported that patients had long-term survival (still alive after the median follow-up time of 45 months) [12]. Patients who were treated with doxorubicin and ifosfamide had a partial response [17]. 

Interestingly, in the case of metastatic and unresectable BML and LG-ESS, hormonal therapy can be considered as both tumors are hormone sensitive and show estrogen and PR expression [32]. In the present case, treatment with oral letrozole (2.5 mg once daily) showed a durable partial response with no adverse events for 28 months.

Because of the rarity of LG-ESS, the clinical course of the disease has not been clearly characterized. It has been known that the period from initial curative surgery to recurrence varies widely and may even exceed 20 years (Table 1). Although recurrent and metastatic LG-ESS showed an excellent 5-year survival (93%) and long-term survival with hormonal therapy [11,12,20], to the best of our knowledge, the natural course of LG-ESS with distant metastasis without treatment has not been reported. Our patient was asymptomatic and lived well for 15 years after the pulmonary lesions were detected. She received letrozole only after ESS was diagnosed 2 years ago. Although this is only one case, it can be assumed that LG-ESS can be event free for a considerable period without any treatment, even with distant metastasis. The patient is satisfied with the current treatment and completely understands the disease status and treatment.

## 4. Conclusions

This patient’s metastatic LG-ESS responded well to hormone therapy and showed a very indolent course for a considerable period of more than 15 years, even without any treatment. In this case, it was possible to correct the diagnosis based on the clinician’s recommendation. Therefore, when a clinical pattern different from that already known is encountered, the possibility of diagnosis re-evaluation should be considered, especially in rare diseases.

## Figures and Tables

**Figure 1 diagnostics-12-00271-f001:**
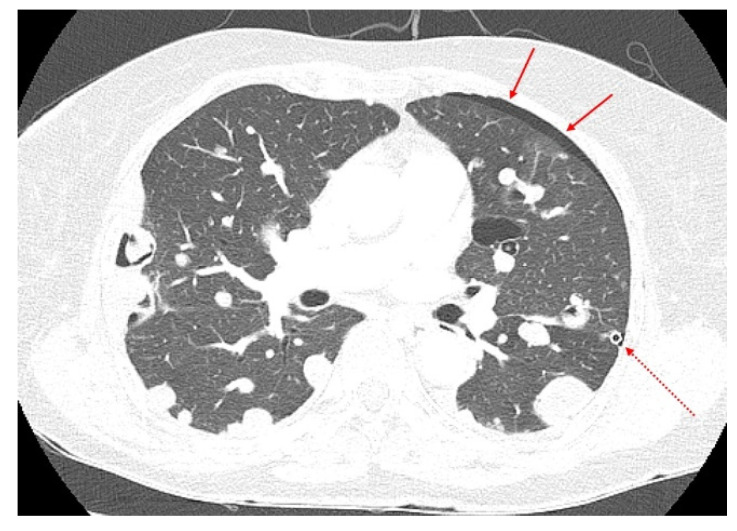
A chest computed tomography (CT) features of the lung masses. A chest CT (April 2017) showed pneumothorax in the left lung and multiple masses in both lungs. The amount of pneumothorax was small (solid arrows) because CT was performed after the chest tube (dotted arrows) was inserted and stabilized.

**Figure 2 diagnostics-12-00271-f002:**
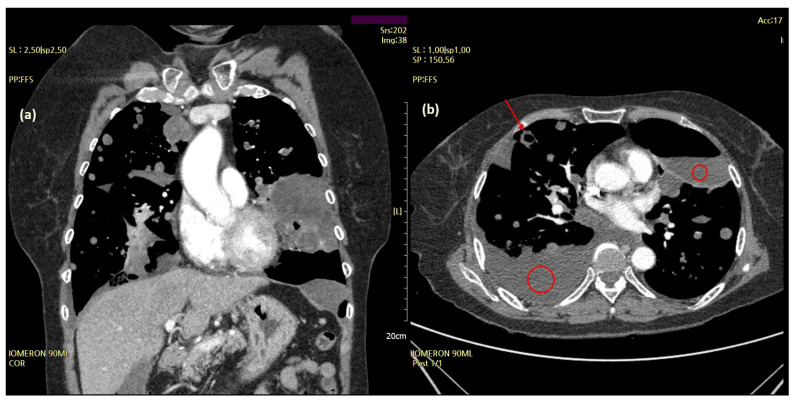
Chest computed tomography (CT) features of the aggravating lesions. Cystic masses (arrow) and large and multiple hydropneumothoraces with massive pleural effusion (red circles) on chest CT (June 2019). (**a**) Coronal and (**b**) axial views.

**Figure 3 diagnostics-12-00271-f003:**
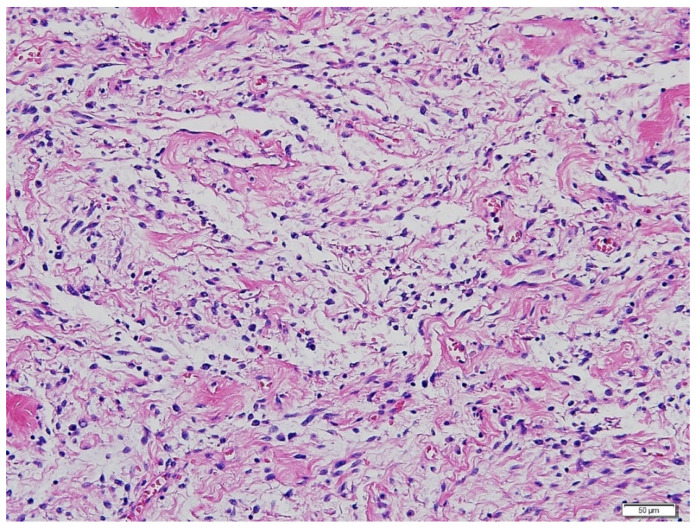
Hematoxylin-eosin staining of the lung mass. Microscopic examination revealed metastatic lung nodules with small, uniform, and bland tumor cells with spindled nuclei and scant cytoplasm, in the background of rich small arterioles or capillary networks. (Magnification 200×).

**Figure 4 diagnostics-12-00271-f004:**
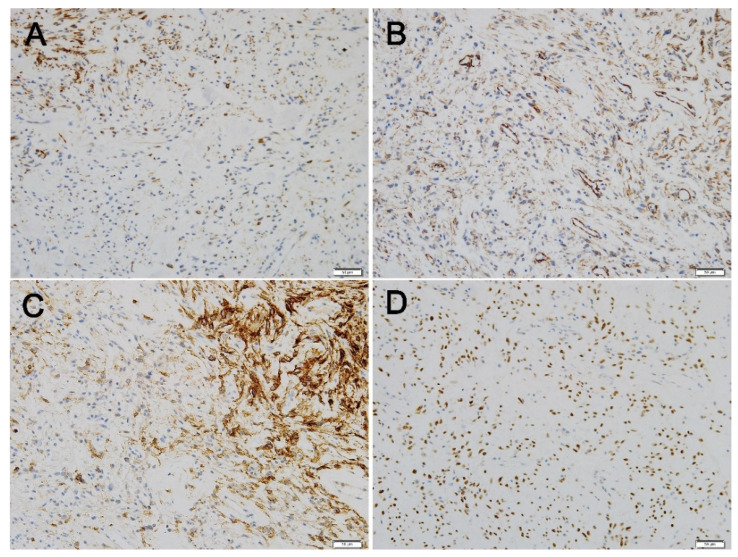
Immunohistochemical staining of the lung mass. (**A**) Desmin is expressed in some tumor cells (200×), (**B**) h-Caldesmon is diffusely positive (200×). (**C**) Tumor cells reveal patchy cytoplasmic immunoreactivity on CD10 (200×). (**D**) Diffuse nuclear positivity of tumor cells with WT1 (200×).

**Figure 5 diagnostics-12-00271-f005:**
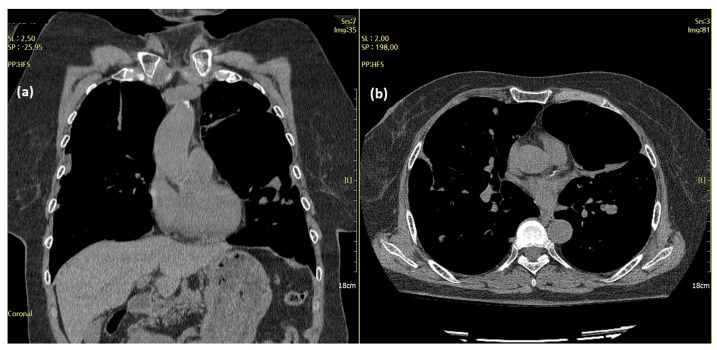
Follow-up computed tomography examination. Follow-up chest computed tomography showed improved disease status (August 2021). (**a**) Coronal and (**b**) axial views.

## Data Availability

Not applicable. Since this manuscript is a case report, there are no available datasets or materials to be shared publicly or available repositories for these data. However, the authors will readily respond and provide the patient’s clinical information upon request, as long as the patient’s personal information is not disclosed.

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
