# Peer review of "Pulmonary Metastasizing Low-Grade Endometrial Stromal Sarcoma: Case Report and Review of Diagnostic Pitfalls"

_diagnostics, 2022, doi:10.3390/diagnostics12020271_

Round 1
Reviewer 1 Report
Dear Authors,
I read with great interest the Case Report titled “Low-grade endometrial stromal sarcoma misdiagnosed as pulmonary benign metastasizing leiomyoma followed by re-diagnosis led by a clinician based on the aggressive clinical features: A case report”.
The authors reported a single case of lung recurrence of Low-grade endometrial stromal sarcoma (LG-ESS), a rare uterine tumor. In particular, a 70-year-old woman, with a history of hysterectomy for uterine leiomyoma 23 years prior, presented with multiple pulmonary nodules. Initially, the biopsy revealed that the pulmonary masses were consistent with benign metastasizing leiomyoma (BML); but subsequently, considering that the size and distribution of the pulmonary masses were aggravated, the gynecologic pathologist reviewed the biopsy. The pathologist confirmed that the correct diagnosis was LG-ESS. The patient started an antiestrogen therapy with Letrozolo and she presented with no recurrence of symptoms or progressive disease.
Comments:
- Lines 115-120: The authors defined only the immunohistochemical
The molecular knowledge on endometrial stromal neoplasms has been rapidly increasing and is considered complementary to morphologic and immunohistochemical findings for better categorization of these tumors. Considering this last evidence, it’s important for the readers that the authors define better the metastatic lesion with molecular classification to be in line with the latest literature. For example, low-grade endometrial stromal sarcoma with sex cord-like differentiation.
- It would be helpful for the reader if the authors would add a table presenting a review of all cases in literature about LG-ESSs recurrence. In fact, LG-ESSs represent a tumor with low malignant potential, often with indolent clinical behavior and with some cases presented with a late recurrence after hysterectomy. In this way, the authors should explain all types of presentations and management of this rare uterine tumor.
- Lines 148-153: The authors described the reasons for the misdiagnosis of BML. Considering the difficulty of diagnosis of LG-ESS with the necessity to review the biopsy by a pathologist; another reason could be represented by an incorrect diagnosis at the first surgery. In fact, the introduction of immunohistochemical is recent. In fact, the last classification of the World Health Organization (WHO) in 2020 sub-categorised ESTs into four groups: Endometrial Stromal Nodule (ESN), Low-Grade Endometrial Stromal Sarcoma (LG-ESS), High-Grade Endometrial Stromal Sarcoma (HG-ESS), and Undifferentiated Uterine Sarcoma (UUS).
- The discussion is somewhat confusing and needs to be edited. It would benefit from review by a native English-speaking person
Best regards
Reviewer 2 Report
Only about 30 cases of pulmonary metastasizng LGESS have been detailed described so far. The further evidence from larger elaborations is usually limited to the information about incidence and localization of metatases. Therefore the manuscript is of intrest. The submitted report is carefully written and thoroughly elaborated in regard to differential diagnosis between benign metatasizing leiomyoma versus pulmonary metastasizing LGESS.
However, some important information is lacking and the discussion can (and
should) be improved.
Since the remaining text is of good quality, the recommendation below should help to elevate the good quality of the manuscript up to an excellent level.
Case Report:
1. In the ovarectomized patient, the rapid growth of both the presumed
leiomyoma or finally LGESS coud be due to estrogen excess (intrinsic or
extrinsic):
a) did the patient received estrogens as HRT?
b) What was the BMI of the patient?
2. Were dyspnea and cough the only symptoms in this patients? Was the pelvic mass fully asymptomatic?
3. Is a more detailed medical history of the patient avaiable (other neoplasms, tamoxifen treatemnt, irradiation history etc.)
4. Were the option of a surgical removal of the pelvic mass and (e.g.
thoracoscopic) resection of the pulmonary oligometastases considered after
initial response to aromatase inhibitors?
5. Were the histological specimens from the initial surgery (23 years ago)
available for a pathological re-review?
6. The identifying information (exact date of examination or surgery) should
be removed.
Discussion:
a) Some closely correspoding reports concerning delayed presentation of pulmonry metastases of LGESS after presumed benign histology are lacking:
- Sanneh A et al. Low-Grade Endometrial Stromal Sarcoma Diagnosed 8 Years After Hysterectomy With Morcellation. Obstet Gynecol. 2020 PMID: 32649495.
- Altal OF et al. Complete remission of advanced low-grade endometrial
stromal sarcoma after aromatase inhibitor therapy: a case report. J Med Case
Rep. 2021 PMID: 33947445;
b) The authors discuss thoroughly the differential diagnosis between benign
metastasizing leiomyoma and metastasizing LGESS (line 177-197). In contrast, the diagnostic and therapeutic pitfalls of pulmonary metastasizing LGESS are
not discussed. However, in presence of pulmonary metastases of
a mesenchymal tumor of genital origin, especially the HGESS can be a
challenge, when misdiagosed at initial surgery. I recommend the reading of
the following recent publications:
- Kostov S et al. A. New Aspects of Sarcomas of Uterine Corpus-A Brief
Narrative Review. Clin Pract. 2021 PMID: 34842646
- Akaev I et al. Update on Endometrial Stromal Tumours of the Uterus.
Diagnostics (Basel). 2021 PMID: 33802452;
- Micci F et al. Molecular pathogenesis and prognostication of "low-grade''
and "high-grade" endometrial stromal sarcoma. Genes Chromosomes Cancer. 2021
PMID: 33099834
c) The paper would benefit from a table summarizing the available evidence
about pulmonary metastasizing LGESS (about 15 publications listed in PubMed, the most of them cited in:
- Mindiola-Romero AE et al. 2021. Metastatic low-grade endometrial stromal
sarcoma after 24 years: A case report and review of recent molecular
genetics. Diagn Cytopathol. 2021 PMID: 32910526.
An older overview accumulating 16 cases was published in 2002:
- Aubry MC et al. Endometrial stromal sarcoma metastatic to the lung: a
detailed analysis of 16 patients. Am J Surg Pathol. 2002 PMID: 11914621.
d) Other treatement options and rationale should be mentioned. For instance, the prerequisite for aromatase inhibitors is the positivity for estrogen receptor (it can be absent in up to 20% of LG-ESS); In case of negativity for ER, but positivity for only progesterone receptor, the progestagens offer a therapeutic option. NCCN guidelines also recommend to consider progestagens (in LG-ESS, megestrol acetate between 80-160 mg daily would be the first choice, and medoroxyprogesterone acetate a less useful alternative (cave contraindications (diabetes) and side effects (thromboembolism).
Round 2
Reviewer 1 Report
Dear Authors,
I read with great interest the modified manuscript titled “Revised diagnosis of low-grade endometrial stromal sarcoma initially misdiagnosed as pulmonary benign metastasizing leiomyoma led by a clinician based on aggressive clinical features: A case report”.
The authors specified that they did not perform a suitable molecular test for LG-ESS, such as JAZF1 translocation, molecular alterations because the molecular analysis on endometrial stromal neoplasms could not be evaluated in their hospital.
The authors added a table presenting a review of all cases in literature about LG-ESSs recurrence.
The authors revised the paragraph in the discussion section about the reasons for the misdiagnosis of BML and the last classification of the World Health Organization (WHO) in 2020 of sub-categorised ESTs into four groups: Endometrial Stromal Nodule (ESN), Low-Grade Endometrial Stromal Sarcoma (LG-ESS), High-Grade Endometrial Stromal Sarcoma (HG-ESS), and Undifferentiated Uterine Sarcoma (UUS).
The authors performed an English language editing.
For the abovementioned considerations I think the paper is much improved.
Best regards
Reviewer 2 Report
I congratulate the Authors to the excellent job they made. The improved manuscript is informative and comprehensive. I recommend to accept it for publication. My last suggestions is, to shorten the title nd to add the "literature review" to the "case report". No additional review round is necessary, my suggestions are:
"Pulmonary metastasizing low-grade endometrial stromal sarcoma: case report and an updated literature review" or
"Pulmonary metastasizing low-grade endometrial stromal sarcoma: case report and review of diagnostic pitfalls".
